# Can Knowledge Graphs Make Large Language Models More Trustworthy? An Empirical Study over Open-ended Question Answering

## Abstract

Recent works integrating Knowledge Graphs (KGs) have led to promising improvements in enhancing reasoning accuracy of Large Language Models (LLMs). However, current benchmarks mainly focus on closed tasks, leaving a gap in the assessment of more complex, real-world scenarios. This gap has also obscured the evaluation of KGs' potential to mitigate the problem of hallucination in LLMs. To fill the gap, we introduce OKGQA, a new benchmark specifically designed to assess LLMs enhanced with KGs under open-ended, real-world question answering scenarios. OKGQA is designed to closely reflect the complexities of practical applications using questions from different types, and incorporates specific metrics to measure both the reduction in hallucinations and the enhancement in reasoning capabilities. To consider the scenario in which KGs may have varying levels of mistakes, we further propose another experiment setting OKGQA-P to assess model performance when the semantics and structure of KGs are deliberately perturbed and contaminated. OKGQA aims to (1) explore whether KGs can make LLMs more trustworthy in an open-ended setting, and (2) conduct a comparative analysis to shed light on methods and future directions for leveraging KGs to reduce LLMs' hallucination. We believe that this study can facilitate a more complete performance comparison and encourage continuous improvement in integrating KGs with LLMs. The code of this paper is released at https://anonymous.4open.science/r/OKGQA-CBB0.

## 1 Introduction

Contemporary LLMs are prone to producing hallucinations due to gaps in their knowledge (Gekhman et al., 2024; Lee et al., 2023), primarily when the training data contain misinformation, biases, or inaccuracies. These flaws can arise when we simply maximize the log-likelihood, leading to responses that may seem plausible, but are often irrelevant or incorrect (Weng, 2024). This issue is especially concerning in scenarios where accuracy and reliability are essential, such as healthcare (He et al., 2023) and science (Taylor et al., 2022).

To address this critical limitation, researchers employ diverse strategies to augment the LLMs by integrating external knowledge graphs (KGs) (Pan et al., 2024; Luo et al., 2023a; Hu et al., 2023; Sun et al., 2023; van Breugel et al., 2021). KGs offer structured, explicit, and up-to-date factual knowledge, including domain-specific knowledge, providing a faithful knowledge source for reasoning (Zheng et al., 2023; Agrawal et al., 2023). Moreover, each piece of information in KGs can be traced back to its source, providing context and provenance. This traceability not only aids in verifying the reliability of the information but also provides clear pathways of reasoning, making the interpretation process transparent. Due to their reliability and interpretability, KGs are considered a promising method to improve the reliability of LLM reasoning.

However, current benchmarks for testing the capabilities of these LLM+KG models are predominantly closed-ended, restricting responses to a limited set of entities/relations (Talmor et al., 2019; Mihaylov et al., 2018; Jin et al., 2020) or a set of logical forms (Yih et al., 2016; Talmor & Berant, 2018; Puerto et al., 2023) derived from specific facts of KG. Hence, they can only test a very limited subset of the LLM's tendency to hallucinate, leaving a gap in the assessment of complex, real-world scenarios.

Particularly, standard metrics such as FActScore (Min et al., 2023) and SAFE (Wei et al., 2024) for evaluating the hallucination rate of LLMs require open-ended settings, *i.e.*, questions are phrased as a statement which requires a longer answer.

**Motivation**: Recognizing the research gap, our study aims to contribute to the existing knowledge from two perspectives: 1) explore whether KGs can make LLMs more trustworthy in an open-ended setting, and 2) conduct a comparative analysis to shed light on methods and directions for leveraging KGs to reduce LLMs' hallucination.

To achieve this, we develop a new benchmark, **O**pen-ended **K**nowledge-**G**raphs **Q**uestion **A**nswering (OKGQA), specifically designed to assess LLMs enhanced with KGs under open-ended, real-world question answering scenarios. OKGQA is designed to closely reflect the complexities of practical applications using questions of different types, and incorporates specific metrics to measure both the reduction in hallucinations and enhancements in reasoning capabilities. Specifically, OKGQA covers 850 queries from ten categories, representing different levels of real-world QA types. All queries are synthesized by LLMs with pre-defined templates mentioned in Table 1, and we use Wikipedia as references for all the entities mentioned in the query. All the questions are designed to be open-ended and cannot be simply answered by generating logical forms or retrieving specific facts from the knowledge graph. To consider the scenarios in which KGs may have varying levels of mistakes (*i.e.*, attributes may be mislabeled, relations may not exist, etc.), we propose another experiment setup OKGQA-P (discussed in Section 3.2) to assess model performance under conditions where KGs' semantics and structure are deliberately perturbed and contaminated. We assess the hallucination ratio and the quality of the response (detailed metrics can be found in Section 3.3) of the tested models.

Based on our experiments, we find that (1) retrieving KG information can indeed mitigate factual errors in LLMs' responses, especially for queries requiring additional reasoning; (2) directly performing reasoning in the LLM itself (e.g., internal reasoning strategies like Chain-of-thoughts (Kim et al., 2023) and self-consistency (Wang et al., 2022)) may cause bias and hallucination; (3) Subgraphs generally achieve the best performance across different query types, especially for simpler types of queries; (4) Integrating KGs can effectively reduce hallucinations in LLMs, even when KGs are contaminated to a certain degree. Overall, our main contributions include:

- We present OKGQA, a benchmark tailored for evaluating LLMs enhanced with KGs under open-ended, real-world question-answering scenarios. The benchmark extends the assessment of closed-ended question answering to an open-ended setting, which can further support the assessment of hallucination of LLMs.

- To consider the scenarios in which KGs may have varying levels of mistakes, we propose another experiment setup OKGQA-P to assess model performance under conditions in which the semantics and structure of KGs are deliberately perturbed and contaminated.

- We conduct a series of experiments on OKGQA and OKGQA-P, analyzing the effectiveness of various retrieval methods and LLMs of different magnitudes, providing insights for future research and development. Our results underscore the importance of integrating LLMs with KGs to help reduce hallucinations, even in circumstances where KGs are contaminated.

## 2 RELATED WORK

Due to the stochastic decoding process of Large language models (LLMs), *i.e.*, sampling the next token in the sequence, LLMs exhibit probabilistic behavior, potentially yielding varied outputs of the same input across different instances (Agrawal et al., 2023). In addition, they also face challenges in accurately interpreting phrases or terms when the context is vague and resides in a knowledge gap region of the model, leading to outputs that may sound plausible but are often irrelevant or incorrect. This "hallucinations" undermines the reliability of LLMs (Huang et al., 2023). One emerging research trend is enhancing LLMs through integrating external knowledge graphs (Agrawal et al., 2023). KGs offer structured, explicit, and up-to-date factual knowledge, including domain-specific knowledge, providing a faithful knowledge source for reasoning (Zheng et al., 2023; Agrawal et al., 2023). Moreover, each piece of information in KGs can be traced back to its source, providing context and provenance. This traceability not only aids in verifying the reliability of the information but also provides clear pathways of reasoning, making the interpretation process transparent. Due to their

reliability and interpretability, KGs are considered a promising method to improve the reliability of LLM reasoning.

Researchers employ diverse strategies to augment the LLMs by integrating external KGs. For example, KAPING (Baek et al., 2023b) matches entities in questions to retrieve related triples from knowledge graphs for zero-shot question answering. Wu et al. (2023) finds that converting these triples into textualized statements can further enhance LLM performance. StructGPT (Jiang et al., 2023b) propose to convert user query into structured formats (e.g., SPARQL) for information extraction from KGs. Following the succuess of internal reasoning-enhancement methods like Chain-of-thoughts (CoT) (Wei et al., 2022), Reflexion (Shinn et al., 2024), and Tree-of-thoughts (ToT), He et al. (2022) propose "rethinking with retrieval" to use decomposed reasoning steps from CoT prompting to retrieve external knowledge, leading to more accurate and faithful explanations. IR-CoT (Trivedi et al., 2022) interleaves the generation of CoT with knowledge retrieval from corresponding KGs, iteratively guiding both retrieval and reasoning for multi-step questions. MindMap (Wen et al., 2023) introduce a plug-and-play approach to evoke graph-of-thoughts reasoning in LLMs. Similarly, Reasoning on Graphs (RoG) (Luo et al., 2023b) use KGs to create faithful reasoning paths based on various relations, enabling interpretable and accurate reasoning in LLMs.

However, current benchmarks for testing the capabilities of these LLM+KG models are predominantly closed-ended, restricting responses to a limited set of entities/relations or a set of logical forms derived from specific facts of KG. Hence, they can only test a very limited subset of the LLM's tendency to hallucinate, leaving a gap in the assessment of complex, real-world scenarios. Particularly, standard metrics such as FActScore (Min et al., 2023) and SAFE (Wei et al., 2024) for evaluating the hallucination rate of LLMs require open-ended settings, *i.e.*, questions are phrased as a statement which requires a longer answer. Compared with previous works, our proposed OKGQA is tailored for evaluating LLMs enhanced with KGs under open-ended, real-world question-answering scenarios. The benchmark extends the assessment of closed-ended question answering to an open-ended setting, which can further support the assessment of hallucination of LLMs.

## 3 OKGQA: A OPEN-ENDED KNOWLEDGE GRAPH QUESTION-ANSWERING BENCHMARK

OKGQA is a benchmark designed to assess LLMs enhanced with KGs under open-ended, real-world questions answering scenarios. OKGQA is designed to closely reflect the complexities of practical applications using questions of different types and incorporates specific metrics to measure both the reduction in hallucinations and enhancement in reasoning capabilities. In this section, we discuss the construction of the dataset, and the construction of OKGQA-P which is a variant of OKGQA where the KGs' semantics and structure are deliberately perturbed and contaminated.

**Motivation of Open-ended QA instead of Close-ended QA**: Current benchmarks for testing the capabilities of LLM+KGs models are predominantly close-ended[1], which demand a short answer such as 'yes' and 'no'. This restricts the responses to a limited set of entities/relations (Talmor et al., 2019; Mihaylov et al., 2018; Jin et al., 2020) or a set of logical forms (Yih et al., 2016; Talmor & Berant, 2018; Puerto et al., 2023) derived from specific KG facts. Hence, they can only test a very limited subset of the LLM's tendency to hallucinate, leaving a gap in the assessment of more complex, real-world scenarios. Particularly, standard metrics (like FActScore (Min et al., 2023) and SAFE (Wei et al., 2024)) for evaluating the hallucination rate of LLMs require open-ended settings, *i.e.*, questions are phrased as a statement which requires a longer answer.

**Open-ended QA Task Definition**: The tasks we design require to first *understand* the scope of the question, then optionally *retrieve* relevant information from multiple parts of the knowledge graph, then finally *synthesize* a coherent and informative response. The ideal output should be a paragraph that fully addresses the question with accurate and factual responses. We verify the response based on the metrics specified in Section 5 to reflect the models' capabilities and faithfulness.

---

[1]https://en.wikipedia.org/wiki/Open-ended_question#Examples

| Statistics (on average) | |
|---|---|
| Tokens in query | 23.97 |
| Total number of queries | 850 |
| Number of unique DBPedia entities | 816 |

| Before Pruning → After PPR Pruning | |
|---|---|
| Tokens in subgraph | 348,715 → 2,452 |
| Number of nodes | 7,171 → 48 |
| Number of Edges | 8,213 → 152 |
| Avg. Degree | 1.15 → 3.17 |
| Clustering Coefficient | 0.00 → 0.69 |
| Graph Density | 0.00 → 0.07 |

| Query Type | Simple | Complex |
|---|---|---|
| Descriptive | 78 | 11 |
| Explanatory | 195 | 56 |
| Predictive | 110 | 55 |
| Comparative | 72 | 74 |
| Critical | 182 | 17 |
| Total | 637 | 213 |

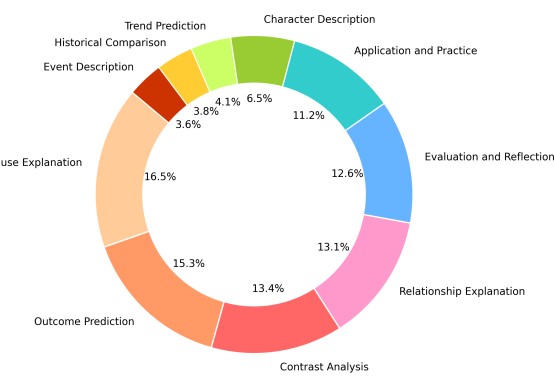

(a) Dataset statistics and query types      (b) Distribution of sub-query types

Figure 1: (left) Dataset statistics and query types, (right) Sub-query type distribution

## 3.1 DATASET CONSTRUCTION

**Queries.** We utilize a template-based methodology to systematically generate a diverse range of queries using LLMs, including categories such as descriptive, explanatory, predictive, comparative, and critical queries. Details regarding the specific templates and example queries can be found in Table 1, while the corresponding prompts are provided in the Appendix A.1. To ensure that the generated queries accurately represent real-world scenarios and complexities, we integrate the corresponding Wikipedia pages of each entity as contextual information for the LLMs' generation. Furthermore, we follow a human-in-the-loop process to enhance the instructions for generating (the details are given in Appendix A.4). Intuitively, starting with an initial instruction, we generate a large number of query candidates, then use an LLM as automatic evaluator to evaluate the quality of the query, denoted as a set of scores from different metrics $s_{auto}$ ranging from 1 to 10, with higher scores indicating better performance. Then, these queries are manually evaluated and assign a set of human-label score $s_{human}$ correspond to $s_{auto}$, normalized as the same range as $s_{auto}$. We then optimize the input instructions by iteratively generate instructions to minimize the gap between $s_{human}$ and $s_{auto}$. These queries are categorized by difficulty and the naturalness of their phrasing. The statistics of the queries can be found in Figure 1.

Table 1: Query types and examples in OKGQA. **Brown** is used to highlight the placeholders (*e.g.*, [person], [event]) in description, while **Teal** refers to the specific entities to replace the placeholders. The distribution of the sub-query types can be referred to in Figure 1b.

| Type | Sub-Type | Description / Template | Example |
|---|---|---|---|
| Descriptive | Character Description | Describe a [person]'s significant contributions during their career. | Please describe **Albert Einstein**'s contributions to the field of **physics**. |
| | Event Description | Provide a detailed description of the background and course of an [event]. | Please provide a detailed description of the background and course of the **French Revolution**. |
| Explanatory | Cause Explanation | Why did [person] take [action] at [time]? | Why did **Nixon** choose to **resign** from the presidency in **1974**? |
| | Relationship Explanation | Explain the relationship between [entity A] and [entity B] and its significance. | Explain the relationship between **Alexander the Great** and **Aristotle** and its significance. |
| Predictive | Trend Prediction | Based on the historical behavior of [entity], what do you think it might do in the future? | Based on **Tesla**'s historical behavior, in which fields do you think it might innovate in the future? |
| | Outcome Prediction | Based on the current situation, how do you predict [event] will develop? | Based on the current international situation, how do you predict **climate change policies** will develop? |
| Comparative | Contrast Analysis | Compare and contrast the similarities and differences between [entity A] and [entity B] in [aspect]. | Compare and contrast the leadership styles of **Steve Jobs** and **Bill Gates**. |
| | Historical Comparison | Compare the impact of [historical event A] and [historical event B]. | Compare the impact of **World War I** and **World War II** on the global order. |
| Critical | Evaluation and Reflection | How do you evaluate the impact of [person/event] on [field]? Please explain your viewpoint. | How do you evaluate **Martin Luther King**'s impact on the **civil rights movement**? Please explain your viewpoint. |
| | Application and Practice | How do you think [theory/method] can be applied to [practical issue]? | How do you think **machine learning technology** can be applied to **medical diagnostics**? |

**KG Sub-graphs.** To reduce the size of KGs, following previous works (Luo et al., 2023a; Yih et al., 2016; Talmor & Berant, 2018), we construct subgraphs of DBpedia (all the queries in OKGQA can

be reasoned based on DBPedia[2]) by extracting all triples contained within the $k$-hop neighbors from the question entities in the query. For our experiments, we set $k$=2. To further reduce the size of the 2-hop subgraphs, we leverage personalized page-rank (PPR) (Bahmani et al., 2010) to prune the nodes/edges that are not relevant to the query (the details of the PPR algorithm are discussed in Appendix A.5). We compare the statistics of subgraphs before and after PPR pruning in Figure 1a.

## 3.2 OKGQA-P: Benchmark with Noise & Perturbations in KGs

To further mimic the real situations where KGs may not be of high quality (*i.e.*, attributes of nodes/edges may be mislabeled, relations may not exist, *etc.*), we propose another experiment setting OKGQA-P in this section to assess the model performance under conditions where KGs' semantics and structure are deliberately perturbed and contaminated. Considering that KGs are typically annotated by humans and are generally accurate and meaningful, we introduce perturbations to edges in the KG to degrade the quality of the KGs, diminishing human comprehensibility. To quantify the degree of perturbation, we evaluate both the semantic and structural similarity between the original and the modified KG as defined below.

**Notation.** Let $\mathcal{F}_\theta$ be a KG-augmented model, and KG as $\mathcal{G} = (\mathcal{V}, \mathcal{E}, \mathcal{T})$, where $\mathcal{V}$ is the set of entities (nodes), $\mathcal{E}$ is the set of relation types (edges), and $\mathcal{T} = \{(v_1, e, v_2)|v_1, v_2 \in \mathcal{V}, e \in \mathcal{E}\}$ is the set of triplets composed from existing entities and relations. Let $\mathcal{G}' = (\mathcal{V}, \mathcal{E}', \mathcal{T}')$ be the KG obtained after perturbing $\mathcal{G}$, where $\mathcal{E}' \neq \mathcal{E}$ and $\mathcal{T}' \neq \mathcal{T}$. Let $f(\mathcal{G}, \mathcal{G}')$ be a function that measures the similarity between $\mathcal{G}$ and $\mathcal{G}'$. Let $g(\mathcal{G})$ be the downstream performance when evaluating $\mathcal{F}_\theta$ on data samples $X$ and $\mathcal{G}$.

**High-level Procedure.** First, we test $\mathcal{F}_\theta$ on data samples $X$ and $\mathcal{G}$ to get the original performance $g(\mathcal{G})$. Second, we perturb $\mathcal{G}$ to obtain $\mathcal{G}'$. Third, we evaluate $\mathcal{F}_\theta$ on data samples $X$ and $\mathcal{G}'$ to get the perturbed performance $g(\mathcal{G}')$. Finally, we measure $g(\mathcal{G}) - g(\mathcal{G}')$ and $f(\mathcal{G}, \mathcal{G}')$ to assess how robust $\mathcal{F}_\theta$ is, *i.e.*, to assess the model performance under conditions where KGs' semantics and structure are deliberately perturbed.

To quantify how much the perturbed KG has deviated from the original KG, *i.e.*, $f(\mathcal{G}, \mathcal{G}')$, we leverage metrics from Raman et al. (2020) for capturing semantics (ATS) and structural (SC2D, SD2) similarity between KGs. Intuitively, ATS leverages an pre-trained LM for link prediction to measure the probability of each edge from $\mathcal{G}'$ existing in $\mathcal{G}$, while SC2D and SD2 measures the structural similarity between two KGs based on local clustering coefficient and degree distribution. For each of the three metrics, higher value indicates higher similarity. The detailed description can be found in Appendix A.7. The results after perturbation can be found in Figure 4.

For the perturbation methods, we consider four perturbation heuristics based on (Raman et al., 2020) as follows: **Relation Swapping (RS)** randomly chooses two edges from $\mathcal{T}$ and swaps their relations. **Relation Replacement (RR)** randomly chooses an edge $v_1, e, v_2 \in \mathcal{T}$, then replaces $e_1$ with another relation $e_2 = \text{argmin}_{r \in \mathcal{R}} S_{\mathcal{G}}(v_1, e, v_2)$, where $S_{\mathcal{G}}(v_1, e, v_2)$ uses ATS to measure the semantics similarity between two edges. **Edge Rewiring (ER)** randomly chooses an edge $(v_1, e, v_2) \in \mathcal{T}$, then replaces $v_2$ with another entity $v_3 \in \mathcal{E} \backslash \mathcal{N}_1(v_1)$, where $\mathcal{N}_1(v_1)$ represents the 1-hop neighborhood of $v_1$. **Edge Deletion (ED)** randomly chooses an edge $(v_1, e, v_2) \in \mathcal{T}$ and deletes it. We control the perturbation level based on the percentage of KG edges being perturbed.

## 3.3 Evaluation Metrics

To quantify the hallucinations of LLMs, we leverage two public metrics, **FActScore** (Min et al., 2023) (factual precision in atomicity score) and **SAFE** (Wei et al., 2024) (search-augmented factuality evaluator). **FActScore** is designed to measure factual precision in text by decomposing a long form generation into multiple atomic facts and validates each separately against a reliable knowledge base, such as Wikipedia. We measure the proportion of facts that are supported by the knowledge source out of the total atomic facts. **SAFE**, on the other hand, adopts a more dynamic approach to fact-checking. It employs a language model as an investigative agent, which iteratively uses Google Search queries and reasons about whether the search results support or do not support the fact.

---

[2]https://www.dbpedia.org/resources/knowledge-graphs/

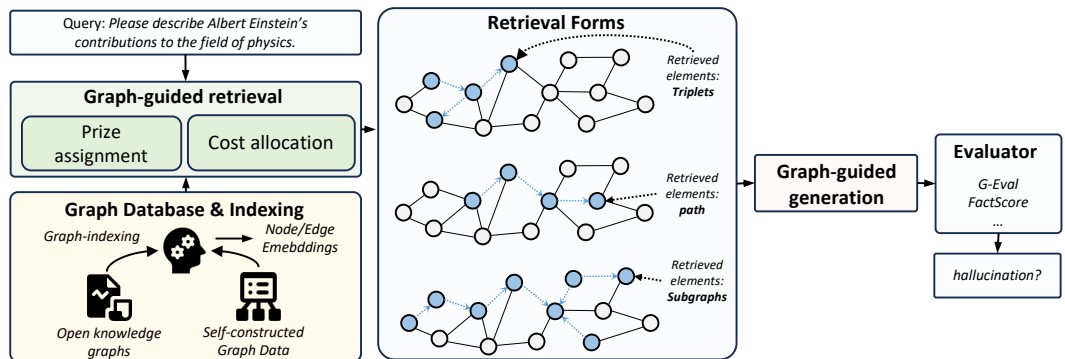

Figure 2: Overview of KG-augmented framework for reducing hallucinations.

In addition to the hallucination metrics, we have also included four metrics, following previous works (Edge et al., 2024; Wang et al., 2023) to quantify the desirable qualities for generating sensible responses. Given the complexities inherent in our open-ended QA setting, collecting and annotating ground-truth answers pose significant challenges. Therefore, we leverage LLMs as automatic reference-free evaluators to assess these metrics based on G-Eval (Liu et al., 2023), a framework of using LLMs with chain-of-thought (CoT) and a form-filling paradigm. To enhance G-Eval's robustness and stability, we provide the relevant Wikipedia pages as context in the prompt when calculating the metrics. The metrics are defined as follows: **Context Relevance**: measures how well the generated response aligns with the provided context. **Comprehensiveness**: evaluates the extent to which the generated answer covers all aspects and details of the question. **Correctness**: measures how clearly and specifically the generated answer responds to the question. **Empowerment**: evaluates how well the generated answer helps the reader understand the topic and make informed decisions. The detailed prompt for each metric can be found in Appendix A.1.

**Models.** We incorporate a range of widely used LLMs of different scales, including GPT-4o[3], GPT-4o-mini[4], Llama-3.1-8B-instruct (Dubey et al., 2024), Mistral-7B-instruct-v0.3 (Jiang et al., 2023a), and Gemma-2-9B-it (Team et al., 2024). We use the embedding model (text-embedding-3-small) from OpenAI for the G-retrieval process, considering cost and performance.

## 4 EXPLORING KG-AUGMENTED FRAMEWORK FOR REDUCING HALLUCINATION

**Overview.** To explore whether KG-augmented approaches can mitigate LLMs' hallucination, we investigated a range of methods and discovered that they can be encapsulated within a unified framework as shown in Figure 2. Our framework follows the paradigm of retrieval augmented generation (RAG) (Edge et al., 2024; Baek et al., 2023a), which retrieves essential information from the KGs, and then uses the retrieved knowledge to enhance the LLM's generation. It has two components, *i.e.*, *Graph-guided retrieval (G-retrieval)* and *Graph-guided generation (G-generator)*, with a variety of algorithmic choices that combine to create diverse algorithms with different benefits for reducing hallucinations. We analyze our proposed strategies within each component in Section 5, aiming to shed light on the best practices for leveraging KGs for reducing hallucinations in LLMs.

Here we formalize the problem of context-relevant KG retrieval for open-ended question answering. Given a user query $q$, a generative pretrained language models (PLMs) first encodes the input tokens, and then models a conditional distribution $p(a|q)$ to generate an output response $a = [a_1, \ldots, a_T]$. To explore whether KGs can help reduce hallucinations of PLMs, we introduce the *retrieved knowledge* from the KG, denoted $\mathcal{Z} \in \mathcal{G}$, which can take various forms (*e.g.*, triplets, paths, subgraphs, as shown in Figure 2). This retrieved knowledge contains information relevant to the user query, and can either benefit the PLM at its knowledge boundaries (Baek et al., 2023b) or explicitly guide its reasoning process (Luo et al., 2023a; Sun et al., 2023). We explore which forms of knowledge should

---

[3]https://openai.com/index/hello-gpt-4o/
[4]https://openai.com/index/gpt-4o-mini-advancing-cost-efficient-intelligence/

be extracted from KGs to best enhance the PLM. The distribution of the retrieved knowledge $\mathcal{Z}$ is modeled as $p(\mathcal{Z}|q, \mathcal{G})$, leading to a new likelihood for generating responses, expressed as $p(a|q)$.

$$
\begin{aligned}
p(a|q) &= \sum_{\mathcal{Z} \subseteq \mathcal{G}} p_\phi(a|q, \mathcal{Z}) p_\theta(\mathcal{Z}|q, \mathcal{G}) \\
&\approx p_\phi(a|q, \mathcal{Z}^*) p_\theta(\mathcal{Z}^*|q, \mathcal{G}),
\end{aligned}
\tag{1}
$$

where $p_\theta(\mathcal{Z}^*|q, \mathcal{G})$ represents the distribution of retrieving relevant knowledge elements from the KGs, and $p_\phi(a|q, \mathcal{Z}^*)$ represents the distribution of leveraging the retrieved knowledge elements to generate the final responses, parameterized by $\phi$ and $\theta$, where $\mathcal{Z}^*$ is the optimal retrieved knowledge. Due to the potential exponential growth in the number of possible retrieved knowledge with graph size, efficient approximation methods are required. Consequently, the first line of Eq. 1 is approximated using the most probable retrieved knowledge $\mathcal{Z}^*$.

## 4.1 GRAPH-GUIDED RETRIEVAL (G-RETRIEVAL)

The graph-guided retrieval phase focuses on extracting the most relevant elements (*e.g.*, triplets, paths or subgraphs) from KGs to help answer the user query. Initially, the query is transformed into an embedding vector $\mathbf{q} \in \mathbb{R}^d$ using a language model encoder $f_{\text{LM}}(\cdot)$. For each node $v \in \mathcal{V}$ and edge $e \in \mathcal{E}$, their respective embeddings $\mathbf{e}_v$ and $\mathbf{e}_e$ are computed using the same language model to map all the information into the same vector space, and their relevance to the query is quantified using cosine similarity as: $s_v = \frac{\mathbf{q} \cdot \mathbf{e}_v}{\|\mathbf{q}\|\|\mathbf{e}_v\|}$ and $s_e = \frac{\mathbf{q} \cdot \mathbf{e}_e}{\|\mathbf{q}\|\|\mathbf{e}_e\|}$. Next, we identify the most relevant nodes and edges for the query based on the corresponding similarity, to yield a set of 'top-$k$ nodes/edges'.

The goal of G-retrieval is to retrieve the knowledge that encompasses as many relevant nodes and edges as possible, while keeping the size manageable. To this end, we leverage a 'prize' and 'cost' trade-off strategy to guide the retrieval process, as follows: (1) *Prize assignment*: based on the computed similarity scores, we assign prizes to nodes and edges to quantify their relevance to the query. Specifically, we assign the top-$k$ nodes/edges with descending prizes values from $k$ to 1, while nodes and edges outside the top-$k$ receive a prize of 0. Formally: $p_v = \max(0, k - \text{rank}(v) + 1)$ and $p_e = \max(0, k - \text{rank}(e) + 1)$. (2) *Cost allocation*: to manage the retrieved knowledge size, we assign penalties as cost $C_e$ (by default assign to 1) during the expansion of the retrieved paths or subgraphs. These prizes and costs will be used in the retrieval process, by aiming to maximize the prizes while minimizing costs.

**Triplet-retrieval**: the triplet retrieval approach retrieves a fixed number $k$ of triplets with the highest total prize assigned to their respective head entity, relation, and tail entities.

**Path-retrieval**: based on the defined prize assignment and cost allocation, the path-retrieval approach starts from a fix number of $k$ of high-prize nodes to construct sequences $\mathcal{P} = \{v_1, e_1, v_2, \ldots, e_{n-1}, v_n\}$ by aiming to greedily maximize their scores: $S(\mathcal{P}) = \sum_{i=1}^{n} p_{v_i} + \sum_{i=1}^{n-1} p_{e_i} - \sum_{i=1}^{n-1} c_e$. We use a priority queue to iteratively extend paths by selecting the highest-scoring extensions while avoiding cycles and adhering to maximum path length and quantity constraints. Finally, the top-scoring paths are sorted and returned. The details of path-retrieval can be found in Appendix A.6.

**Sub-graph retrieval**: the sub-graph retrieval approach also leverages the prize assignment and cost allocation; we follow He et al. (2024) to optimize the process using Prize-Collecting Steiner Tree (PCST) algorithm. The PCST problem seeks to build a connected subgraph $\mathcal{S}$ that maximizes the total prizes of included nodes and edges while minimizing the total cost of the edges. The score of the subgraph is defined as $S(\mathcal{S}) = \sum_{n \in V_\mathcal{S}} p_{v_i} + \sum_{e \in E_\mathcal{S}} p_{e_i} - \sum_{e \in E_\mathcal{S}} c_e$. Unlike in path-retrieval, we only yield one subgraph that maximizes the total score.

## 4.2 GRAPH-GUIDED GENERATION (G-GENERATOR)

After retrieving $\mathcal{Z}^*$, the graph-guided generation phase focuses on learning to leverage this knowledge to generate the corresponding output to answer the user query. The generation is modeled as a sequential decision-making process as follows, where at each time step $t$, the next token $a_t$ is generated by G-Generator, conditioned on the query $q$, the retrieved knowledge $\mathcal{Z}^*$, and all previously

generated tokens $a_{0:t-1}$.

$$p(a|q, \mathcal{Z}^*) = \prod_{t=1}^{T} p_\theta \left( a_t | q, \mathcal{Z}^*, a_{0:t-1} \right) \qquad (2)$$

The model learns the parameters $\theta$ to maximize the likelihood of generating the correct output sequence by computing the probability distribution over possible next tokens at each time step. This ensures that the generated output not only considers the previously generated tokens but also the retrieved knowledge. The generation continues until a termination condition is met, such as generating an end-of-sequence token or reaching the maximum sequence length $T$.

## 5 EXPERIMENTS

### 5.1 RQ1: MAIN RESULTS - CAN KGS REDUCE HALLUCINATION IN LLMS?

To explore whether KGs can help reduce hallucination in LLMs, we benchmark the LLMs in different settings. We use zero-shot and few-shot prompting as baselines without injecting external knowledge. In addition, compared to adding external knowledge from KGs, we also consider leveraging LLMs' internal knowledge to do Chain-of-thought (Kim et al., 2023), or self-consistency (Wang et al., 2022) prompting as baselines. As for LLMs augmented with KGs, we consider three KG retrieval forms: triplets, paths, and subgraphs (as shown in Figure 2) for comparison, to study the impact of G-retrieval for reducing LLMs' hallucinations. The results can be found in Table 2 and Figure 3. We obtain some intriguing findings:

Table 2: Comparison results of various forms of information extracted from the KGs.

| Models | G-Eval | | | | Hallucination | |
|---|---|---|---|---|---|---|
| | Context Relevance | Comprehensiveness | Correctness | Empowerment | SAFE | FActScore |
| **Baseline: Without External Knowledge (Zero-shot prompting)** | | | | | | |
| GPT-4o | 68.12% ± 0.88% | 65.41% ± 0.79% | 60.41% ± 0.38% | 62.41% ± 0.84% | 82.47% ± 0.62% | 55.34% ± 0.93% |
| GPT-4o-mini | 63.21% ± 0.49% | 60.11% ± 0.47% | 55.43% ± 0.63% | 58.72% ± 0.62% | 80.14% ± 0.89% | 50.23% ± 1.01% |
| llama-3.1-8b-instruct | 57.12% ± 0.91% | 54.74% ± 1.20% | 49.01% ± 0.61% | 52.21% ± 0.71% | 79.33% ± 0.91% | 45.14% ± 0.32% |
| mistral-7B-Instruct-v0.3 | 55.71% ± 1.21% | 52.00% ± 1.31% | 47.03% ± 0.94% | 50.13% ± 1.04% | 78.27% ± 0.83% | 44.37% ± 1.23% |
| gemma-2-9b-it | 53.63% ± 1.33% | 50.00% ± 1.33% | 45.72% ± 0.71% | 48.15% ± 0.93% | 77.11% ± 0.78% | 40.94% ± 0.83% |
| **Baseline: Without External Knowledge (4-shot prompting)** | | | | | | |
| GPT-4o | 70.61% ± 0.62% | 67.43% ± 0.81% | 62.33% ± 0.37% | 64.51% ± 0.12% | 83.39% ± 0.53% | 57.45% ± 0.78% |
| GPT-4o-mini | 65.53% ± 0.94% | 62.33% ± 1.03% | 57.23% ± 0.68% | 60.47% ± 0.83% | 81.62% ± 0.69% | 52.34% ± 0.76% |
| llama-3.1-8b-instruct | 59.43% ± 0.32% | 56.31% ± 0.78% | 51.27% ± 0.32% | 54.33% ± 0.41% | 80.27% ± 0.78% | 47.24% ± 1.03% |
| mistral-7B-Instruct-v0.3 | 57.34% ± 1.04% | 54.13% ± 1.31% | 49.27% ± 0.84% | 52.46% ± 0.94% | 79.12% ± 0.87% | 45.13% ± 1.42% |
| gemma-2-9b-it | 55.24% ± 1.49% | 52.27% ± 1.21% | 47.14% ± 0.36% | 50.36% ± 0.51% | 78.00% ± 0.77% | 44.32% ± 1.58% |
| **Var-1: With CoT Prompting** | | | | | | |
| GPT-4o - CoT | 72.76% ± 0.92% | 69.56% ± 0.74% | 64.48% ± 0.63% | 66.69% ± 0.69% | 80.07% ± 0.83% | 54.30% ± 0.87% |
| GPT-4o - CoT+SC | 75.81% ± 0.65% | 71.62% ± 0.74% | 66.55% ± 0.75% | 68.74% ± 0.15% | 79.03% ± 0.48% | 53.23% ± 0.78% |
| llama-3.1-8b-instruct - CoT | 61.54% ± 0.95% | 58.35% ± 1.05% | 53.31% ± 0.71% | 56.42% ± 0.83% | 77.07% ± 0.85% | 46.15% ± 0.54% |
| llama-3.1-8b-instruct - CoT+SC | 63.69% ± 0.32% | 60.44% ± 0.59% | 55.46% ± 0.52% | 58.53% ± 1.11% | 76.00% ± 0.63% | 45.05% ± 0.97% |
| mistral-7B-Instruct-v0.3 - CoT | 59.58% ± 0.43% | 56.23% ± 2.31% | 51.28% ± 1.31% | 54.33% ± 0.72% | 75.04% ± 0.95% | 43.03% ± 1.03% |
| mistral-7B-Instruct-v0.3 - CoT+SC | 61.35% ± 0.93% | 58.33% ± 1.02% | 53.42% ± 0.79% | 56.47% ± 0.85% | 74.30% ± 0.21% | 42.00% ± 0.29% |
| gemma-2-9b-it - CoT | 57.34% ± 1.05% | 54.12% ± 0.32% | 49.27% ± 0.85% | 52.12% ± 0.95% | 72.07% ± 1.05% | 40.13% ± 0.49% |
| gemma-2-9b-it - CoT+SC | 59.42% ± 0.27% | 56.27% ± 0.84% | 51.34% ± 1.42% | 54.34% ± 1.31% | 71.09% ± 0.43% | 39.85% ± 1.03% |
| **Var-2: With Triplets Extracted from KGs Provided** | | | | | | |
| GPT-4o | 74.62% ± 0.65% | 70.44% ± 0.79% | 65.37% ± 0.72% | 67.12% ± 0.71% | 89.20% ± 1.42% | 72.53% ± 0.83% |
| GPT-4o-mini | 69.50% ± 0.81% | 65.03% ± 0.92% | 60.21% ± 0.65% | 63.43% ± 1.01% | 87.52% ± 0.34% | 67.73% ± 0.95% |
| llama-3.1-8b-instruct | 63.45% ± 1.13% | 59.33% ± 1.05% | 54.23% ± 0.75% | 57.33% ± 0.12% | 85.37% ± 0.72% | 62.37% ± 0.82% |
| mistral-7B-Instruct-v0.3 | 61.34% ± 0.31% | 57.21% ± 0.89% | 52.29% ± 0.32% | 55.12% ± 0.43% | 84.21% ± 0.84% | 60.28% ± 1.05% |
| gemma-2-9b-it | 59.25% ± 1.06% | 55.29% ± 0.44% | 50.15% ± 0.85% | 53.73% ± 0.95% | 83.18% ± 0.43% | 58.13% ± 0.91% |
| GPT-4o - CoT+SC | 76.71% ± 0.53% | 72.34% ± 0.21% | 67.33% ± 1.31% | 69.64% ± 0.33% | 88.11% ± 0.57% | 71.45% ± 0.53% |
| **Var-3: With Paths Extracted from KGs Provided** | | | | | | |
| GPT-4o | 78.71% ± 0.53% | 74.53% ± 0.31% | 69.42% ± 0.23% | 71.63% ± 0.61% | 90.20% ± 0.59% | 75.61% ± 0.51% |
| GPT-4o-mini | 73.64% ± 0.93% | 69.41% ± 0.22% | 64.35% ± 0.72% | 67.52% ± 0.82% | 88.22% ± 0.34% | 70.53% ± 0.24% |
| llama-3.1-8b-instruct | 67.51% ± 0.46% | 63.62% ± 1.39% | 58.41% ± 0.93% | 61.57% ± 0.94% | 86.33% ± 0.94% | 65.42% ± 0.95% |
| mistral-7B-Instruct-v0.3 | 65.48% ± 0.94% | 61.37% ± 1.01% | 56.34% ± 0.23% | 59.45% ± 0.43% | 85.26% ± 0.85% | 63.31% ± 1.33% |
| gemma-2-9b-it | 63.35% ± 1.37% | 59.23% ± 0.91% | 54.31% ± 0.91% | 57.41% ± 0.27% | 84.13% ± 0.21% | 61.23% ± 1.04% |
| GPT-4o - CoT+SC | 80.87% ± 0.42% | 76.60% ± 0.65% | 71.54% ± 0.53% | 73.79% ± 1.21% | 89.11% ± 0.63% | 74.53% ± 0.24% |
| **Var-4: With Subgraphs Extracted from KGs Provided** | | | | | | |
| GPT-4o | 80.81% ± 0.43% | 76.63% ± 0.65% | 71.57% ± 0.51% | 73.70% ± 0.62% | 90.83% ± 0.63% | 75.33% ± 0.29% |
| GPT-4o-mini | 75.70% ± 0.44% | 71.51% ± 0.83% | 66.43% ± 0.76% | 69.60% ± 0.65% | 88.71% ± 0.72% | 70.12% ± 0.87% |
| llama-3.1-8b-instruct | 69.61% ± 0.84% | 65.45% ± 0.93% | 60.41% ± 0.65% | 63.42% ± 0.45% | 86.12% ± 0.35% | 65.44% ± 0.87% |
| mistral-7B-Instruct-v0.3 | 67.55% ± 0.87% | 63.35% ± 0.43% | 58.37% ± 0.71% | 61.45% ± 0.32% | 85.21% ± 0.81% | 63.12% ± 0.94% |
| gemma-2-9b-it | 65.45% ± 0.95% | 61.23% ± 1.0% | 56.31% ± 0.35% | 59.40% ± 0.85% | 84.51% ± 0.99% | 63.74% ± 0.49% |
| GPT-4o - CoT+SC | 82.90% ± 0.57% | 78.72% ± 0.61% | 73.64% ± 0.43% | 75.80% ± 0.75% | 89.12% ± 0.94% | 75.42% ± 1.31% |

**Retrieving KG information can indeed mitigate factual errors in the responses.** Methods integrating knowledge extracted from KGs show clear improvements in factual accuracy and comprehension scores compared to the baselines. For example, under Var-2 (triplet retrieval), GPT-4o achieves a

FActScore of 72.55% ± 0.85%, which is a significant increase over the baseline score of 55.35% ±0.95%. Moreover, these methods can be combined with strategies like CoT+SC, enhancing response quality with minimal increase in hallucination ratio. The radar chart in Figure 3 further emphasizes that in most query types, integrating knowledge retrieved from KGs mitigates the hallucination issue compared to baselines, particularly in query types such as "Evaluation and Reflection," "Outcome Prediction," and "Cause Explanation," which require more reasoning and analysis rather than merely listing information. The findings also apply to open-source models like mistral-7B-Instruct-v0.3 and Llama-3.1-8B-instruct, illustrating the consistency of reliability of integrating KGs in LLM reasoning.

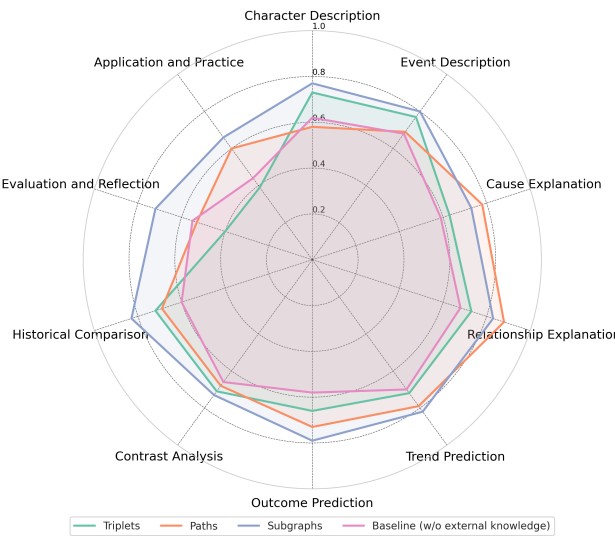

Figure 3: Comparison results of different forms of information over different queries.

**Directly performing reasoning in the LLM itself does not mitigate hallucinations.** We benchmark the hallucination ratio of LLMs using internal reasoning strategies like Chain-of-thought and Self-consistency, as shown in Var-1 in Table 2. It shows that these methods can indeed improve response quality (i.e., correctness, comprehensiveness and empowerment) compared to baselines, but do not consistently improve factuality, and sometimes even diminish factuality. This showcases that relying solely on internal reasoning does not mitigate hallucination, emphasizing the importance of external knowledge to counter the hallucination issue.

**Subgraph retrieval generally achieves best performance across different query types, especially for simpler queries.** We demonstrate the performance of different retrieval methods across different query types in Figure 3, showing that subgraphs generally achieve best performance. Especially for simpler queries ("Character Description" and "Event Description" which generally don't require intensive reasoning), subgraph retrieval methods achieves better performance compared to paths and triplets. Even for queries like "Relationship Explanation" and "Cause Explanation" which require step-wise reasoning, subgraph methods still demonstrate promising performance. This suggests that while different forms of retrieved knowledge each offer unique benefits suitable for specific types of queries, subgraphs provide consistently good performance broadly.

## 5.2 RQ2: HOW ARE KG-AWARE METHODS AFFECTED BY NOISE / PERTURBATIONS IN KGS?

We benchmark different KG-augmented LLMs on our OKGQA-P setting, where we deliberately perturb and contaminate the semantics and structure of KGs to simulate the real-world situation where KGs may not have high quality. Specifically, we consider different perturbation methods discussed in Section 3.2 and control the perturbation level based on the percentage of KG edges being perturbed. We first illustrate how much the perturbed KG has been deviated from the original KG with the increase of perturbation level, shown in Figure 4.

It shows that the perturbation methods like edge deletion, rewiring and swapping have relatively weak influence on ATS (which intuitively measures semantic similarity), even as the perturbation level increases. For the edge deletion methods, only if the perturbation level reaches 1.0, the ATS goes to

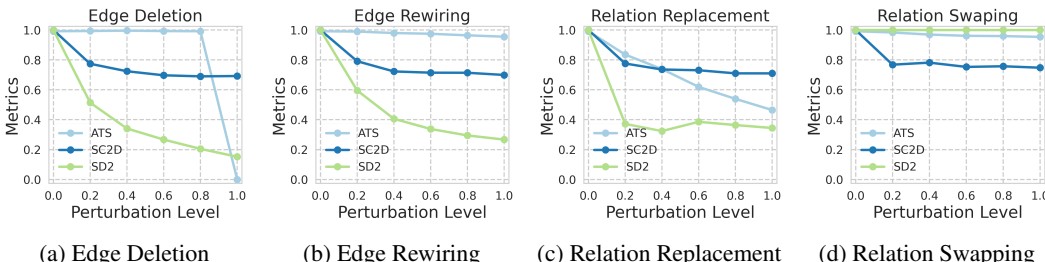

Figure 4: Performance Metrics (ATS, SC2D, SD2) vs. Perturbation Level for Different Perturbation Methods.

0, otherwise, the ATS is very high. Relation replacement has a relatively strong impact on ATS, i.e., on the semantics of the perturbed KGs. For SC2D and SD2 (which intuitively measure structural similarity), the four methods show similar levels of decrease in most cases.

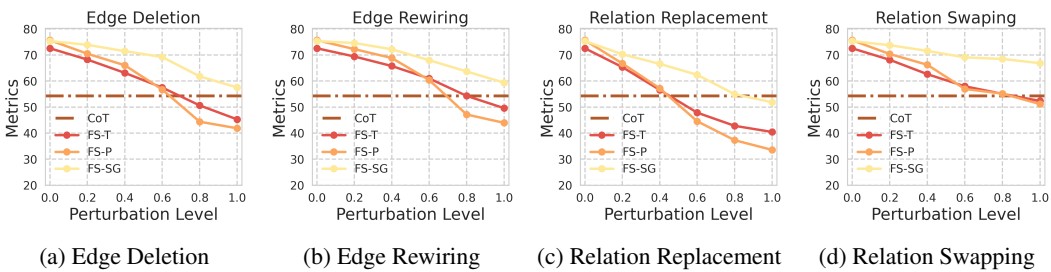

Figure 5: Performance Metric (FActScore) vs. Perturbation Level for Different Perturbation Methods and Different Retrieval Methods. **FS-T** refers to FActScore metric using triplets, **FS-P** refers to using paths, and **FS-SG** refers to using sub-graphs.

Figure 5 illustrates the hallucination ratio using different methods on the perturbed KGs. We observe that (1) FS-SG consistently outperforms FS-T and FS-P even at higher perturbation levels, demonstrating its robustness by maintaining higher scores as perturbations increase. (2) FS-T and FS-P exhibit similar trends, each showing a significant performance drop as perturbation levels increase. Particularly, performance of FS-T and FS-P deteriorate when the perturbation level reaches 50%, *i.e.*, becoming worse than the baseline using CoT. (3) On the setting using Relation Replacement which severely harms the semantics of the KGs, FS-T and FS-P decline more sharply than FS-SG. However, it still outperforms the baseline when the perturbation level is smaller than 40%.

In conclusion, our study demonstrates that leveraging external knowledge from KGs can effectively reduce hallucinations in LLMs, even when KGs are contaminated to a certain degree. In real-world applications, KGs such as Wikidata and unlikely to experience severe perturbations due to their continuous updates and community-driven quality control, making our findings applicable in most real-world scenarios.

## 6 CONCLUSION

In this paper, we present OKGQA, a benchmark tailored for evaluating LLMs enhanced with KGs under open-ended, real-world question answering scenarios. The benchmark extends the assessment from close-ended question answering to the open-ended setting, to support the assessment of hallucination in LLMs. To further mimic real-world scenarios where KGs may not have high quality, we propose another experiment setup OKGQA-P to assess model performance under conditions where KG's semantics and structure are deliberately perturbed and contaminated. We conduct a series of experiments on OKGQA and OKGQA-P, analyzing the effectiveness of various retrieval methods and LLMs of different magnitudes, providing insights for further research and development. Our results underscore the significance of integrating KGs with LLMs to help reduce hallucination of LLMs, even in circumstances where the KGs are contaminated.

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

## A  APPENDIX

### A.1  PROMPT LIST

In this section, we present all the prompts required for the main experiments. To enhance clarity, we provide only one example in the prompt labeled as Example 1; the other few-shot examples utilized are labeled as Other In-Context Few-shots within the prompt.

### A.1.1 KNOWLEDGE-AUGMENTED LANGUAGE MODEL PROMPTING

**System Instruction:** "You are a helpful assistant designed to answer the users' open-ended questions. Your task is to provide accurate, concise, and useful information to foster understanding and solve problems. Whether the questions relate to complex scientific concepts, historical events, practical advice, or everyday life, your goal is to assist by offering thoughtful and informative responses."

Example 1:

**Question**: How do you think blockchain technology can be applied to secure online voting systems? Please provide specific suggestions?

Below are the facts that might be relevant to answer the question: ('Bitcoin', 'wikiPageWikiLink', 'Bill_Gates'), ('Bitcoin', 'wikiPageWikiLink', 'Non-fungible_token'), ('Decentralized_application', 'wikiPageWikiLink', 'Cardano_(blockchain_platform)'), ('Non-fungible_token', 'wikiPageWikiLink', 'Decentralized_application'), ('Hyperledger', 'wikiPageWikiLink', 'Blockchain'), ('Blockchain', 'implementations', 'Bitcoin'), ('Blockchain', 'currencyName', 'Litecoin'), ('Thing', 'type', 'Privacy_and_blockchain'), ('Thing', 'type', 'Saraju_Mohanty'), ('Thing', 'type', 'Blockchain'), ('Thing', 'type', 'Satoshi_Nakamoto'), ('Thing', 'type', 'Pseudonym'), ('Thing', 'type', 'Metaverse'), ('Thing', 'type', 'PricewaterhouseCoopers')

**Answer**: Blockchain technology can significantly enhance the security of online voting systems through its decentralized, transparent, and immutable nature. By distributing data across multiple nodes, blockchain eliminates single points of failure, reducing the risk of hacking or vote tampering. The system allows for verifiable, yet anonymous, vote recording through cryptographic methods, ensuring voter privacy while preventing fraud such as double voting. Additionally, once recorded, votes cannot be altered, safeguarding election integrity. Smart contracts can further automate the voting process, enabling instant and accurate tallying, while the system's resilience to cyberattacks like DDoS ensures uninterrupted functionality.

Other In-Context Few-shots

**Question**: {Question}

**Below are the facts that might be relevant to answer the question:** {Retrieved-knowledge}

**Answer**:

### A.1.2 OKGQA QUERY GENERATION PROMPTING

**System Instruction:** Generate open-ended questions about different types: character description, event description, cause explanation, relationship explanation, trend prediction, outcome prediction, contrast analysis, historical comparison, evaluation and reflection, and application and practice.

Some templates are provided below: character description: describe a [person]'s significant contributions during their career. Example: Please describe Albert Einstein's contributions to the field of physics. event description: Provide a detailed description of the background and course of an [event]. Example: Please provide a detailed description of the background and course of the French Revolution. cause explanation: Why did [person] take [action] at [time]? Example: Why did Nixon choose to resign from the presidency in 1974? relationship explanation: Explain the relationship between [entity A] and [entity B] and its significance. Example: Explain the relationship between Alexander the Great and Aristotle and its significance. trend prediction: Based on the historical behavior of [entity], what do you think it might do in the future? Example: Based on Tesla's historical behavior, in which fields do you think it might innovate in the future? outcome prediction: Based on the current situation, how do you predict [event] will develop? Example: Based on the current international situation, how do you predict climate change policies will develop? contrast analysis: Compare and contrast the similarities and differences between [entity A] and [entity B] in [aspect]. Example: Compare and contrast the leadership styles of Steve Jobs and Bill Gates. historical comparison: Compare the impact of [historical event A] and [historical event B]. Example: Compare the impact of World War I and World War II on the global order evaluation and reflection: How do you evaluate the impact of [person/event] on [field]? Please explain your viewpoint. Example: How do you evaluate Martin Luther King's impact on the civil rights movement? Please explain your viewpoint. application and practice: How do you think [theory/method] can be applied to [practical issue]?

**Listing 1** Example 1

```
1   {    'question': """Compare and contrast the similarities
2           and differences between the Apple iPhone and
3           Samsung Galaxy in terms of user interface design.',
4       'type': 'contrast analysis',
5       'placeholders': {
6           'entity A': 'Apple iPhone',
7           'entity B': 'Samsung Galaxy',
8           'aspect': 'user interface design'
9       },
10      'naturalness': 'high',
11      'difficulty': 'medium',
12      'dbpedia_entities': {
13          'entity A': 'http://dbpedia.org/resource/IPhone',
14          'entity B': 'http://dbpedia.org/resource/Samsung_Galaxy'
15      }
16  }
```

Please provide specific suggestions. Example: How do you think machine learning technology can be applied to medical diagnostics? Please provide specific suggestions. Generate the questions, the type of the questions, the placeholders, the naturalness of your generated questions (choose from high, medium, and unnatural), the difficulty of the generated questions (choose from hard, medium and easy) and dbpedia_entities (link the placeholders to dbpedia entities) in JSON format.

Example 1: as shown in Listing 1.

Other In-Context Few-shots

**Generation**:

A.2   PROMPTS FOR INSTRUCTION TUNER

Act as an "Instruction Tuner" for the LLM, you will be given the inputs: (1) the {Current Instruction} used to guide the LLMs's evaluation, including specific examples with ground truth labels; (2) {Current Errors} that emerged with this instruction was applied to the dataset.

The current errors are presented in the following format: (1) INPUT: {input text} (2) PREDICTED OUTPUT: {predicted label}, (3) EXPECTED OUTPUT: {ground truth label}. Carefully analyze these errors and craft a revised concise instruction for the LLM to fit the expected outputs. Include 2-3 examples at the end of your response to demonstrate how the new instruction would be applied.

A.3   METRICS PROMPT FOR G-EVAL

**System Instruction:** "You are a helpful assistant designed to evaluate the quality of the response to a query. Your task is to rate the response on one metric defined as below:"

Empowerment Criteria: Evaluate whether the 'Actual Output' can help the reader understand the topic and make informed decisions regarding the 'Input'. A response with high empowerment provides accurate information and explanations that enhance the reader's understanding. When evaluating empowerment, consider the relevance of the information provided in the 'Actual Output' to the 'Input' and the 'Retrieval Context'.

Comprehensiveness Criteria: Evalute the extent to which the 'Actual Output' covers all aspects and details of the question 'Input'. A comprehensive answer should thoroughly address every part of the question, leaving no important points unaddressed. When evaluating comprehensiveness, consider the relevance of the information provided in the 'Actual Output' to the 'Input' and the 'Retrieval Context'.

`Correctness Criteria`: Measure how clearly and specifically the 'Actual output' responds to the question 'input'. A highly direct response stays focused on the question, providing clear and unambiguous information. When evaluating correctness, consider the relevance of the information provided in the 'Actual Output' to the 'Input' and the 'Retrieval Context'.

`Context Relevance Criteria`: Evaluate the extent to which the 'Actual output' incorporates relevant information from the 'Retrieval Context'. This includes assessing whether the output adheres to the thematic, factual, and situational specifics presented in the 'Retrieval Context'. Relevant responses not only address the direct query but also align closely with the contextual elements provided, ensuring a seamless and coherent transition between the 'Retrieval Context' and the 'Actual Output'. The most contextually relevant responses demonstrate an understanding and appropriate reflection of the given circumstances, historical facts, or conceptual background, thereby contributing to the overall accuracy and utility of the information provided.

### A.4 QUERY CONSTRUCTION

In this section, we discuss the details of the query construction of OKGQA. It follows a template-based method with LLMs to generate a diverse range of queries. To ensure that the generated queries accurately represent real-world scenarios and complexities, we integrate the corresponding Wikipedia pages of each entity as contextual information for the LLMs' generation. Furthermore, we follow a human-in-the-loop process to optimize the instruction used for generation, as shown in Figure 6. Intuitively, starting with an initial instruction, we generate a large number of query candidates, then use an LLM as automatic evaluator to evaluate the quality of the query, denoted as a set of scores from different metrics $s_{\text{auto}}$ ranging from 1 to 10, with higher scores indicating better performance. Then, these queries are manually evaluated and assign a set of human-label score $s_{\text{human}}$ correspond to $s_{\text{auto}}$, normalized as the same range as $s_{\text{auto}}$. We then optimize the input instructions by iteratively generate instructions to minimize the gap between $s_{\text{human}}$ and $s_{\text{auto}}$. This process is quite mimic the way of reinforcement learning with human feedback (RLHF) (Ouyang et al., 2022) and inherits the benefit that labeling the reward or penalty of the LLMs' output is much easier than labeling the output directly.

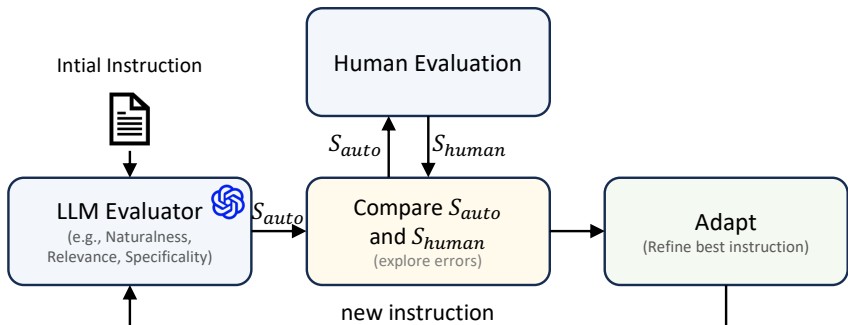

Figure 6: Process of human-in-the-loop of query construction.

Specifically, we consider five angles to measure the quality of the generated queries: (1) **Naturalness**: assessing how fluid and human-like the query sounds; (2) **Relevance**: measuring whether the query pertains directly to the entity and the context provided; (3) **Specificity**: determining the level of detail and granularity included in the query, ensuring it is not too broad or vague; (4) **Novelty**: evaluating the uniqueness of the query, ensuring it is not just a repetitive or common question; (5) **Actionability**: gauging whether the query prompts clear, definite answers or actions that are feasible within the given context. Each of these angles contributes to a holistic evaluation of the query's effectiveness and relevance in real-world applications.

We have three evaluators participating in the manual assessment of query quality. All of the evaluators are computer science majors with fluent English skills. As the evaluation centers on various linguistic metrics such as naturalness, relevance, specificity, novelty, and actionability, we only require the evaluators to possess a fundamental understanding of English without restricting their majors. We calculate the Pearson correlation coefficients between human and LLM scores as shown in Table 3.

Table 3: Pearson correlation coefficients between human and LLM scores across rounds.

| Metric | Round 1 | Round 2 | Round 3 | Round 4 |
|---|---|---|---|---|
| Naturalness | 0.60 | 0.65 | 0.69 | 0.74 |
| Relevance | 0.55 | 0.59 | 0.64 | 0.70 |
| Specificity | 0.46 | 0.54 | 0.60 | 0.65 |
| Novelty | 0.49 | 0.57 | 0.63 | 0.67 |
| Actionability | 0.33 | 0.41 | 0.48 | 0.53 |

Table 4: Cohen's Kappa coefficient for various metrics.

| Metric | Evaluator 1 & 2 | Evaluator 1 & 3 | Evaluator 2 & 3 |
|---|---|---|---|
| Naturalness | 0.85 | 0.83 | 0.84 |
| Relevance | 0.81 | 0.79 | 0.80 |
| Specificity | 0.65 | 0.63 | 0.66 |
| Novelty | 0.60 | 0.58 | 0.61 |
| Actionability | 0.67 | 0.65 | 0.68 |

It shows that as the rounds progress, agreement between humans and LLMs increases, suggesting that iterative feedback improves alignment between human annotation and LLMs response.

In addition, we also consider to verify the inter-rater reliability across three evaluators as shown in Table 4. We report the Cohen's Kappa coefficient for each two evaluators as follows. Using the Landis & Koch (1977) interpretation guidelines, the Cohen's Kappa coefficients for Naturalness and Relevance (ranging from 0.79 to 0.85) fall within the "Substantial" to "Almost Perfect" categories, indicating strong inter-rater reliability for these metrics. This reflects a shared understanding of the evaluation criteria, resulting in consistent ratings among evaluators. For Specificity, Novelty, and Actionability, the coefficients range from 0.58 to 0.68, placing them primarily in the "Moderate" to "Substantial" categories. These results suggest moderate reliability for these metrics, likely due to subjective interpretation and less clearly defined evaluation guidelines. Novelty, with lower coefficients around 0.61 to 0.63, highlights variability in ratings, suggesting that evaluators may have differing perspectives on what qualifies as novel (but the inter-rater reliability is still be considered "Substantial". Meanwhile, Actionability performs slightly better, nearing the "Substantial" range, indicating moderately consistent criteria.

## A.5 PERSONALIZED PAGERANK (PPR)

In this section, we discuss the details of the PPR algorithm used in Section 3.1 to prune the graph from DBPedia and concentrate on nodes most pertinent to the central nodes of interest. The PPR is calculated using the iterative formula:

$$\mathbf{p} = \alpha \mathbf{A}^\top \mathbf{p} + (1 - \alpha)\mathbf{s}, \tag{3}$$

where $\mathbf{p} \in \mathbb{R}^n$ is the PPR vector representing the relevance scores of $n$ nodes in the graph. $\alpha$ is the damping factor controlling the probability of continuing the random walk versus restarting from the personalization vector. $\mathbf{A}^\top$ is the transpose of the column-normalized adjacency matrix $\mathbf{A}$ of the graph, representing transition probabilities between nodes. $\mathbf{s} \in \mathbb{R}^n$ is the personalization vector, where we assign a value of 1 to the central nodes and 0 to all other nodes to emphasize their importance. To ensure convergence and computational efficiency, we set a tolerance parameter tol $= 1 \times 10^{-6}$ and a maximum iteration limit max_iter $= 100$. After computing the PPR vector $\mathbf{p}$, we apply a threshold of $1 \times 10^{-5}$ to prune the graph. Nodes with PPR scores below this threshold are considered insignificant with respect to the central nodes and are thus removed. This process effectively filters out less relevant nodes, resulting in a pruned graph that highlights the most significant relationships and structures pertinent to our analysis.

A.6    PRIZE-COST-BASED PATH RETRIEVAL

The path-retrieval method is designed to construct and evaluate paths in a graph based on predefined prize assignments and cost allocations. The objective is to form sequences of nodes and edges, represented as $\mathcal{P} = \{v_1, e_1, v_2, \ldots, e_{n-1}, v_n\}$, that maximize the overall score and minimize the costs. To efficiently manage the exploration of potential paths, we utilizes a **priority queue**, a data structure that allows paths to be organized based on their scores, ensuring that the highest-scoring paths are processed first. The method starts by picking a number of starting nodes with high prizes. We then expand these starting points by exploring neighboring nodes. For each neighbor, the method calculates a new score. This score is the sum of the neighbor's prize and the edge's prize minus the edge's cost. If this neighbor hasn't been visited before, which helps avoid looping, the algorithm adds this neighbor to the path. This new path is then added to the priority queue. This expansion keeps going until the path reaches a maximum length or can't be extended further. The algorithm keeps track of paths already explored to avoid repetition and ensure paths don't loop back on themselves. When no more paths can be added or the priority queue is empty, the algorithm sorts the paths by their scores from highest to lowest.

A.7    KG SIMILARITY METRICS

To measure how much the perturbed KGs are deviated from the original KG used in Section 3. We re-present the according metrics from (Raman et al., 2020) below:

**Aggregated Triple Score (ATS)** ATS measures semantic similarity between two KGs. Let $s_{\mathcal{G}}$ be an edge (triple) scoring function, such that $s_{\mathcal{G}}(e_1, r, e_2)$ measures how likely edge $(e_1, r, e_2)$ is to exist in $\mathcal{G}$. Also, assume $s_{\mathcal{G}}$ has been pre-trained on $\mathcal{G}$ for link prediction. Then, ATS is defined as $f_{\text{ATS}}(\mathcal{G}, \mathcal{G}') = \frac{1}{|\mathcal{T}'|} \sum_{(e_1, r, e_2) \in \mathcal{T}'} s_{\mathcal{G}}(e_1, r, e_2) \in [0, 1]$, which denotes the mean $s_{\mathcal{G}}$ score across all edges in $\mathcal{G}'$. Intuitively, if a high percentage of edges in $\mathcal{G}'$ are also likely to exist in $\mathcal{G}$ (i.e., high ATS), then we say that $\mathcal{G}'$ and $\mathcal{G}$ have high semantic similarity. $s_{\mathcal{G}}$ is task-specific, as KGs from different tasks may differ greatly in semantics. We use the $s_{\mathcal{G}}$ from (Li et al., 2016); While ATS captures semantic KG differences, it is not sensitive to KG connectivity structure. Note that $f_{\text{ATS}}(\mathcal{G}, \mathcal{G})$ may not equal 1, since $s_{\mathcal{G}}$ may not perfectly generalize to KGs beyond those it was trained on.

**Similarity in Clustering Coefficient Distribution (SC2D)** SC2D measures structural similarity between two KGs and is derived from the local clustering coefficient (Saramäki et al., 2007; Onnela et al., 2005; Fagiolo, 2007). For a given entity in $\mathcal{G}$ (treated here as undirected), the local clustering coefficient is the fraction of possible triangles through the entity that exist (i.e., how tightly the entity's neighbors cluster around it). For entity $e_i \in \mathcal{E}$, the local clustering coefficient is defined as $c_i = 2\text{Tri}(e_i)/(\deg(e_i)(\deg(e_i) - 1))$, where $\text{Tri}(e_i)$ is the number of triangles through $e_i$, and $\deg(e_i)$ is the degree of $e_i$. For each relation $r \in \mathcal{R}$, let $\mathcal{G}^r$ be the subgraph of $\mathcal{G}$ consisting of all edges in $\mathcal{T}$ with $r$. That is, $\mathcal{G}^r = (\mathcal{E}, r, \mathcal{T}')$, where $\mathcal{T}' = \{(e, r, e') \mid e, e' \in \mathcal{E}\}$. Let $\mathbf{c}^r$ denote the $|\mathcal{E}|$-dimensional clustering coefficient vector for $\mathcal{G}^r$, where the $i$th element of $\mathbf{c}^r$ is $c_i$. Then, the mean clustering coefficient vectors for $\mathcal{G}$ and $\mathcal{G}'$ are $\mathbf{c}_o = \frac{1}{|\mathcal{R}|} \sum_{r \in \mathcal{R}} \mathbf{c}^r$ and $\mathbf{c}_p = \frac{1}{|\mathcal{R}'|} \sum_{r \in \mathcal{R}'} \mathbf{c}^r$, respectively. SC2D is defined as $f_{\text{SC2D}}(\mathcal{G}, \mathcal{G}') = 1 - \frac{\|\mathbf{c}_o - \mathbf{c}_p\|_2}{\|\mathbf{c}_o - \mathbf{c}_p\|_2 + 1} \in [0, 1]$, with higher value indicating higher similarity.

**Similarity in Degree Distribution (SD2)** SD2 also measures structural similarity between two KGs, while addressing SC2D's ineffectiveness when the KGs' entities have tiny local clustering coefficients (e.g., the item KG used by recommender systems is roughly bipartite). In such cases, SC2D is always close to one regardless of perturbation method, thus rendering SC2D useless. Let $\mathbf{d}^r$ denote the $|\mathcal{E}|$-dimensional degree vector for $\mathcal{G}^r$, where the $i$th element of $\mathbf{d}^r$ is $\deg(e_i)$. Then, the mean degree vectors for $\mathcal{G}$ and $\mathcal{G}'$ are $\mathbf{d}_o = \frac{1}{|\mathcal{R}|} \sum_{r \in \mathcal{R}} \mathbf{d}^r$ and $\mathbf{d}_p = \frac{1}{|\mathcal{R}'|} \sum_{r \in \mathcal{R}'} \mathbf{d}^r$, respectively. SD2 is defined as $f_{\text{SD2}}(\mathcal{G}, \mathcal{G}') = 1 - \frac{\|\mathbf{d}_o - \mathbf{d}_p\|_2}{\|\mathbf{d}_o - \mathbf{d}_p\|_2 + 1} \in [0, 1]$, with higher value indicating higher similarity.

A.8    ADDITIONAL LITERATURE REVIEW

Recent advancements in retrieval-augmented generation (RAG) have introduced innovative strategies to mitigate knowledge hallucination in large language models (LLMs). Coarse-to-Fine Highlighting Lv et al. (2024) utilizes a hierarchical filtering process to progressively refine retrieved knowledge, ensuring that only the most relevant and accurate information is used, making it particularly effective for addressing hallucinations in complex queries. SuRe Kim et al. (2024) focuses on summarizing

retrieved documents into concise, query-relevant representations, enhancing both computational efficiency and the contextual alignment of knowledge with the query. Similarly, RECOMP Xu et al. (2023) introduces dynamic context compression and selective augmentation, tailoring retrieved information to the query's specific needs to improve response quality and relevance. Collectively, these methods underscore the importance of balancing retrieval breadth with precision and computational efficiency in RAG systems. Their principles align closely with the goals of integrating knowledge graphs (KGs) to reduce hallucination in LLMs, as explored in OKGQA. By adopting techniques such as hierarchical filtering, summarized retrieval, and adaptive augmentation, future KG-augmented frameworks can enhance the reliability and accuracy of open-ended QA tasks, even when dealing with noisy or incomplete knowledge.

