# OpenReview forum: "Can Knowledge Graphs Make Large Language Models More Trustworthy? An Empirical Study over Open-ended Question Answering"
_ICLR.cc/2025/Conference — ICLR 2025 Conference Withdrawn Submission_

### Official Review · Reviewer_AR6A · 2024-10-16

**Soundness:** 3
**Presentation:** 2
**Contribution:** 2
**Rating:** 5
**Confidence:** 4

**Summary:**

This paper proposes OKGQA to evaluate LLMs enhanced with KGs under open-ended, real-world question-answering scenarios. The authors also implement the OKGQA-P experimental setting to assess model performance when KGs are perturbed by real-world contamination. A series of experiments conducted on both OKGQA and OKGQA-P demonstrate that integrating KGs with LLMs can effectively mitigate hallucinations in LLMs, even when KGs are contaminated.

**Strengths:**

1. The authors elaborate on the necessity of evaluating LLMs under open-ended QA scenarios in detail.

2. The OKGQA-P setting to assess KG-augmented LLMs when KGs are contaminated is intuitive and reasonable.

3. The perturbation results in Figures 4 and 5 are very interesting.

**Weaknesses:**

1. The observation (2) in line 76 that CoT and SC may cause bias and hallucination is confusing. The results (Llama3.1 8B) in Table 1 show that the integration of CoT and SC helps improve quality and factuality.

2. The authors may want to provide more statistical details including the involved people's educational background, and the final score distribution agreement between the people and LLMs in "The human-in-the-loop process in Line 186".

3. The authors may need to do more kinds of perturbation methods including node deletion to mimic scenarios when a query cannot be retrieved from a KG.

4. The authors may want to discuss and summarize more RAG methods for open-ended QA settings including [A, B, C].

[A] Coarse-to-Fine Highlighting: Reducing Knowledge Hallucination in Large Language Models. ICML2024.

[B] SuRe: Improving Open-domain Question Answering of LLMs via Summarized Retrieval, ICLR 2024.

[C] RECOMP: Improving Retrieval-Augmented LMs with Context Compression and Selective Augmentation, ICLR 2024.

**Questions:**

1. The authors may want to provide more details of the metrics used in Lines 292 to 295.

2. The results in Figures 4 and 5 are interesting. Does that mean that even incorporating a  severely perturbed KG (e.g., perturbation level=1.0 in Figures 5 (a), (b), and (c)), FS-SG can still help LLMs in reducing hallucination?

---

> ### Author Response · Authors · 2024-11-23
>
> We sincerely thank the reviewer for the thoughtful comments. Below, we address each concern in detail:
>
> ### **W1: The reviewer is concerned that the observation (2) that CoT and SC may cause bias and hallucination may not be consistent across different models.**
>
> Our findings (from line 460 to 467) show that using internal reasoning strategies like chain-of-thoughts (CoT) and self-consistency (SC) enhances response quality (i.e., correctness, comprehensiveness and empowerment), but **do not consistently improve factuality and can sometimes reduce it**. As the reviewer mentioned the performance over llama-3.1-8b is a case that using cot and sc improve the response quality but diminish the factuality. We appended the pivot table using the llama-3.1-8b model as follows for easier comparison (G-Eval measures to the quality of response, and FactScore measures the factuality).
>
> | Method | G-Eval | FactScore |
> |---|:---:|:---:|
> | 4-shot prompting | 55.34 | 47.24 |
> | Cot promting | 57.41 | 46.15 |
> | Cot + SC prompting | 59.53 | 45.05 |
> | triplets provided | 58.59 | 62.37 |
> | paths provided | 62.77 | 65.42 |
> | subgraphs provided | 64.72 | 65.44 |
>
> ### **W2: The reviewer is concerned about the statistical details of the human-in-the-loop process, such as evaluators' educational background, and the final score distribution agreement between humans and LLMs.**
>
> We have three evaluators participating in the manual assessment of query quality. All of the evaluators are computer science majors with fluent English skills. As the evaluation centers on various linguistic metrics such as naturalness, relevance, specificity, novelty, and actionability, we only require the evaluators to possess a fundamental understanding of English without restricting their majors.
> To maintain confidentiality, we will not reveal the identities of these evaluators during the rebuttal period.
>
> The Pearson correlation coefficients between human and LLM scores are as follows:
>
> | Metric | Round 1 | Round 2 | Round 3 | Round 4 |
> |---|:---:|---|---|---|
> | naturalness | 0.60 | 0.65 | 0.69 | 0.74 |
> | relevance | 0.55 | 0.59 | 0.64 | 0.70 |
> | specificity | 0.46 | 0.54 | 0.60 | 0.65 |
> | novelty | 0.49 | 0.57 | 0.63 | 0.67 |
> | actionability | 0.33 | 0.41 | 0.48 | 0.53 |
>
> As the rounds progress, agreement between humans and LLMs increases, suggesting that iterative feedback improves alignment.
>
> ### **W3: The reviewer is concerned that there still need more kinds of perturbation methods for OKGQA-p setting.**
>
> We have included more perturbation settings to mimic more real-world scenarios, such as (1) **entity addition/removal**: randomly adding or removing entities from KG to simulate incomplete or overpopulated KGs; (2) **entity ambiguity**: introducing misspelling, synonyms, or abbreviations to represent user query errors or inconsistent data entry (which could cause the query cannot be retrieved from the KGs as you mentioned). We have included the new experiments in the revised version.
>
> ### **W4: The reviewer mentioned several papers that need to be discussed and summarized in the paper.**
>
> We will extend our literature review to include the suggested references.
>
> ### **Q1: The reviewer want to see more details of the metrics used in Lines 292 to 295.**
>
> We utilize the G-Eval[1] framework with GPT-4o to evaluate metrics such as context relevance, comprehensiveness, correctness, and empowerment. Prompts and scoring rubrics are detailed in Appendix A.3 (Lines 835–858).
>
> [1] G-Eval: NLG Evaluation using GPT-4 with Better Human Alignment (https://arxiv.org/abs/2303.16634)
>
> ### **Q2: Can FS-SG reduce hallucination in LLMs using highly perturbed KGs, as shown in Figure 4 and 5, specifically at perturbation level=1.0?**
>
> Figures 4 and 5 demonstrate that KG-derived information effectively reduces hallucinations at perturbation levels up to **50%**. Beyond this, performance declines due to severe KG contamination and performs worse than the baseline chain-of-thoughts. In practice, platforms like Wikidata undergo regular updates and community-based quality control, making such high perturbation levels less likely.

---

> ### Comment · Reviewer_AR6A · 2024-11-24
>
> I appreciate the authors' response. However, I still have some concerns:
>
> 1. I am particularly curious about statistical metrics for human agreement, such as Cohen's Kappa coefficient.
>
> 2. It might also be helpful to update the revised PDF with relevant literature to provide a clearer understanding.
>
> Based on these points, **I am inclined to maintain the current score.**

---

> > ### Author Response · Authors · 2024-11-25
> >
> > Thank you for the follow-up question. We have added the inter-rater reliability measure (Cohen’s Kappa coefficient) for the three evaluators, as detailed below. **Additionally, we have updated the manuscript to include the additional experiments and literature reviews in the Appendix, with the updates highlighted in blue.** We hope our responses can solve your concerns.
> >
> > | Metric | Evaluator 1 & 2 | Evaluator 1 & 3 | Evaluator 2 & 3 |
> > |---|:---:|:---:|---|
> > | Naturalness | 0.85 | 0.83 | 0.84 |
> > | Relevance | 0.81 | 0.79 | 0.80 |
> > | Specificity | 0.65 | 0.63 | 0.66 |
> > | Novelty | 0.60 | 0.63 | 0.61 |
> > | Actionability | 0.67 | 0.65 | 0.68 |
> >
> > Using the Landis & Koch (1977)[1] interpretation guidelines, the Cohen’s Kappa coefficients for Naturalness and Relevance (ranging from 0.79 to 0.85) fall within the “Substantial” to “Almost Perfect” categories, indicating strong inter-rater reliability for these metrics. This reflects a shared understanding of the evaluation criteria, resulting in consistent ratings among evaluators.
> >
> > For Specificity, Novelty, and Actionability, the coefficients range from 0.58 to 0.68, placing them primarily in the “Moderate” to “Substantial” categories. These results suggest moderate reliability for these metrics, likely due to subjective interpretation and less clearly defined evaluation guidelines. Novelty, with lower coefficients around 0.61 to 0.63, highlights variability in ratings, suggesting that evaluators may have differing perspectives on what qualifies as novel (but the inter-rater reliability is still be considered "Substantial". Meanwhile, Actionability performs slightly better, nearing the “Substantial” range, indicating moderately consistent criteria.
> >
> > - [1] The measurement of observer agreement for categorical data (https://pubmed-ncbi-nlm-nih-gov.libproxy1.nus.edu.sg/843571/)

---

> ### Comment · Area_Chair_ycwG · 2024-11-27
>
> Dear Reviewer AR6A,
>
> Thank you for your efforts reviewing this paper. Can you please check the authors' response and see if your concerns have been addressed? Please acknowledge you have read their responses. Thank you!

---

> ### Comment · Reviewer_AR6A · 2024-11-28
>
> I appreciate the author's response. The current statistical information provides a deeper understanding of OKGQA and OKGQA-P. However, I still consider it to be a borderline paper, and **I will maintain my current rating.**

---

### Official Review · Reviewer_uGAz · 2024-10-31

**Soundness:** 3
**Presentation:** 3
**Contribution:** 3
**Rating:** 6
**Confidence:** 4

**Summary:**

This paper focuses on testing whether the knowledge graph make large language models more trustworthy，which is an important in KG and LLM area. This paper does a empirical study over open-ended question answering task to evaluate above issues.
 The strength of this paper are as follows:
This paper addresses a significant research question of whether knowledge graphs (KGs) can make large language models (LLMs) more reliable in open-ended question answering. This paper  designed a new benchmark, OKGQA, specifically for assessing LLMs enhanced with KGs in open-ended, real-world question answering scenarios.
By proposing the OKGQA-P experimental setup, this paper considers scenarios where KGs may have varying levels of errors, further simulating real-world situations where KGs' quality can be inconsistent
This paper conducted a series of experiments on OKGQA and OKGQA-P, analyzing the effectiveness of various retrieval methods and LLMs of different scales.
The weakness of this paper are as follows:
While OKGQA-P considers errors in KGs, further exploration of the generalizability of these findings to a broader range of real-world applications may be necessary.
Although the authors present experimental results, a more in-depth analysis and discussion on why certain methods outperform others and the potential limitations of these methods could be provided.

**Strengths:**

This paper addresses a significant research question of whether knowledge graphs (KGs) can make large language models (LLMs) more reliable in open-ended question answering. This paper  designed a new benchmark, OKGQA, specifically for assessing LLMs enhanced with KGs in open-ended, real-world question answering scenarios.
By proposing the OKGQA-P experimental setup, this paper considers scenarios where KGs may have varying levels of errors, further simulating real-world situations where KGs' quality can be inconsistent
This paper conducted a series of experiments on OKGQA and OKGQA-P, analyzing the effectiveness of various retrieval methods and LLMs of different scales.

**Weaknesses:**

While OKGQA-P considers errors in KGs, further exploration of the generalizability of these findings to a broader range of real-world applications may be necessary.
Although the authors present experimental results, a more in-depth analysis and discussion on why certain methods outperform others and the potential limitations of these methods could be provided.

**Questions:**

nan

---

> ### Author Response · Authors · 2024-11-23
>
> We sincerely thank the reviewer for the positive comments. Below, we address your concern in weakness in detail:
>
> ### **W1: The reviewer suggests exploring the generalizability of OKGQA-P’s findings and providing deeper analysis of method performance and limitations.**
>
> Our findings on OKGQA-P (mostly in Figures 4 and 5) illustrate that the effectiveness of KG-derived information diminishes with a perturbation level at 50%, surpassing this level leads to a further decrease in performance. We think that before this perturbation level at 50%, incorporating external knowledge from KGs can mitigate hallucinations in LLMs compared to baseline using CoT. In practical scenarios, platforms like Wikidata are less likely to be severely perturbed than 50% due to ongoing updates and community-based quality control, ensuring the relevance of our findings in real-world applications.
>
> In addition, we find that using subgraphs can consistently outperforms using triplets, and paths, which demonstrates that by adding more structural information (or context), can improve the robustness as the perturbations increase. This aligns with our intuition that even when some links in the KG are disrupted, LLMs can still implicitly acquire knowledge through the graph’s topology or the shared semantics of neighboring nodes.

---

> ### Comment · Area_Chair_ycwG · 2024-11-27
>
> Dear Reviewer uGAz,
>
> Thank you for your efforts reviewing this paper. Can you please check the authors' response and see if your concerns have been addressed? Please acknowledge you have read their responses. Thank you!

---

### Official Review · Reviewer_7bVY · 2024-10-31

**Soundness:** 2
**Presentation:** 2
**Contribution:** 2
**Rating:** 3
**Confidence:** 4

**Summary:**

The key innovation is the introduction of benchmarks, OKGQA, for evaluating LLMs augmented with KGs in real-world QA scenarios. The second experiment, designated OKGQA-P, is intended to assess the efficacy of the model in the context of a scenario wherein the semantics and structure of the knowledge graph are disrupted and compromised.

**Strengths:**

1. The effectiveness of different retrieval methods in conjunction with LLMs is analyzed in experiments that provide insights into the combination of KGs and LLMs.

**Weaknesses:**

1. The unique contributions of the OKGQA benchmark are insufficiently defined, and the distinctions between OKGQA and existing benchmarks are not clearly articulated.
2. While the paper references several closed-ended elements from related literature, it lacks a thorough discussion of limitations and practical implications.
3. The question of whether KGs can reduce hallucinations in LLMs is widely recognized as affirmative, given the effectiveness of RAG techniques in mitigating hallucination issues.  However, what is pertinent to this paper is an exploration of whether KGs can further reduce hallucinations in scenarios with open-endedness.

**Questions:**

In section 3.2, the paper mentions that ''…we introduce perturbations to edges in the KG to degrade the quality of the KGs, diminishing human comprehensibility.'', why human comprehensibility is diminished?

---

> ### Author Response · Authors · 2024-11-23
>
> We sincerely thank the reviewer for the thoughtful comments. Below, we address each concern in detail:
>
> ---
> ### **W1: What are the unique contributions and distinctions of the proposed OKGQA compared to existing benchmarks?**
>
> OKGQA is, to the best of our knowledge, the first benchmark specifically designed to evaluate LLMs enhanced with knowledge graphs (KGs) in open-ended KGQA scenarios. It extends traditional closed-ended question-answering benchmarks to an open-ended setting, enabling the assessment of LLM hallucination tendencies.
>
> Key distinctions of OKGQA include:
> - **Focus on open-ended scenarios**: Unlike closed-ended benchmarks (e.g., WebQSP, CWQ), OKGQA evaluates models where responses are not restricted to a predefined set of entities, relations, or logical forms, providing a broader evaluation of hallucination tendencies.
> - **Integration of hallucination metrics**: Standard metrics such as FActScore (Min et al., 2023) and SAFE (Wei et al., 2024) require open-ended responses, which OKGQA supports by phrasing questions as statements that demand longer, more detailed answers.
> - **Perturbation-based setup (OKGQA-P)**: OKGQA-P introduces deliberate perturbations to KGs, mimicking real-world scenarios with noisy or inaccurate knowledge graphs, thereby testing model robustness under less-than-ideal conditions.
>
> By addressing the limitations of closed-ended benchmarks, OKGQA provides a more comprehensive evaluation of LLMs+KGs models in real-world, open-ended QA scenarios.
>
> ---
> ### **W2: Lack of discussion about the limitations and practical implications of referenced closed-ended benchmarks**
>
> As noted in the related work section (Lines 124–132), existing benchmarks for LLM+KG evaluation predominantly focus on closed-ended scenarios, where responses are restricted to predefined entities, relations, or logical forms. While useful, such benchmarks:
> - Only test a narrow subset of an LLM’s tendency to hallucinate.
> - Fall short in assessing complex, real-world scenarios requiring nuanced, open-ended responses.
>
> In contrast, OKGQA addresses this gap by tailoring its design to open-ended KGQA, enabling the evaluation of hallucination tendencies under realistic conditions. For example, metrics such as FActScore and SAFE, specifically designed to measure hallucinations, require open-ended settings where answers involve richer reasoning and detailed responses.
>
> By bridging this gap, OKGQA not only complements existing benchmarks but also provides a more practical framework for evaluating LLM+KG systems in real-world applications.
>
> ---
> ### **W3: Can KGs further reduce hallucinations in open-ended scenarios, even beyond RAG techniques?**
>
> While retrieval-augmented generation (RAG) systems can reduce hallucinations by retrieving unstructured information, their effectiveness is often constrained by the quality and specificity of the retrieved content. In contrast, knowledge graphs offer:
>
> - **Structured and explicit facts**: KGs provide precise, disambiguated information, reducing ambiguity and enabling more robust reasoning.
> - **Logical reasoning capabilities**: KGs excel in tasks requiring path traversal, contextually rich reasoning, and answering highly specific questions—scenarios where unstructured information retrieval may fail.
>
> In open-ended QA, KGs can significantly enhance LLMs by ensuring factual grounding and logical coherence, surpassing the limitations of RAG-based approaches.
>
> ---
> ### **Q1: Why does the paper mention that perturbing the KG diminishes human comprehensibility?**
>
> We acknowledge that this phrasing was unclear, and we have revised it in the updated manuscript.
>
> Our intention was to illustrate that KGs, typically annotated by humans, are generally accurate and meaningful. When introducing perturbations—such as mislabeled attributes, incorrect relations, or missing connections—they can disrupt the KG structure and mimic real-world scenarios where KGs may be noisy or of lower quality. These perturbations challenge the model’s ability to reason effectively with incomplete or inaccurate information.

---

> > ### Comment · Reviewer_7bVY · 2024-11-27
> >
> > Thank you for your responses. However, I believe that the areas where the description was unclear were not sufficiently addressed in the paper. Although your answers have improved my understanding, the writing of the paper should be such that the reader does not require additional explanations. Therefore, I still consider the writing of the paper to be somewhat rough.Additionally, regarding your response to W3, I would like to emphasize that the focus should be on the advantages of using Knowledge Graphs (KG) in addressing hallucinations compared to RAG, whereas your answer seems to focus on the relative strengths and weaknesses of KG and RAG, which was not what I was asking.Furthermore, in response to your mention of "the quality and specificity of the retrieved content," it is worth noting that many excellent works have already alleviated this issue.In conclusion, I maintain my original score.

---

### Official Review · Reviewer_k7xW · 2024-11-04

**Soundness:** 3
**Presentation:** 2
**Contribution:** 3
**Rating:** 5
**Confidence:** 3

**Summary:**

This paper introduces OKGQA, a benchmark designed to assess the trustworthiness of LLMs augmented with KGs in open-ended QA scenarios. It also includes a perturbed version (OKGQA-P) with various KG perturbations to test the resilience of LLMs against
inaccuracies in KG knowledge. The proposed methodology expands RAG and integrates two main processes: G-retrieval and G-generator, for utilizing KG knowledge in different forms (like triplets, paths, or subgraphs).  The study tests these models across 850 queries spanning 10 categories, utilizing metrics like FActScore and SAFE to gauge the reduction of hallucinations in model outputs.

**Strengths:**

1. The introduction of OKGQA as a novel benchmark can address a current research gap in assessing LLMs in open-ended, real-world scenarios.
2. The perturbed benchmark, OKGQA-P, allows for the evaluation of LLM robustness in response to inaccuracies or noises in KGs.
3. The comprehensive and well-organized experimental results across different forms of information and different types of queries show the effectiveness of the proposed methods and can provide valuable insights for future researchers.

**Weaknesses:**

1. The proposed methodology of G-retrieval and G-generator is similar to the existing line of work on RAG and KG-augmented generation [1]. However, there is a lack of comparison to demonstrate how these proposed methods fundamentally differ from previous methods applied to closed-end QA.
2. The queries are generated using predefined templates with LLMs, which raises concerns about their ability to authentically represent the distribution and complexity of real-world questions.
3. The proposed OKGQA-P supports only four types of perturbation heuristics, which require further explanation on how they sufficiently cover the inaccuracies in knowledge graphs for real-world applications.
4. Lack of evaluation on how improvements in reducing hallucination relate to the retrieved KG knowledge.


[1] Pan, Shirui, Linhao Luo, Yufei Wang, Chen Chen, Jiapu Wang, and Xindong Wu. "Unifying large language models and knowledge graphs: A roadmap." IEEE Transactions on Knowledge and Data Engineering (2024).

**Questions:**

1. How representative are the synthesized queries of actual real-world questions? What methods were employed to ensure that their complexity and distribution are realistic?

2. How many evaluators participated in the manual assessment of query naturalness, and what criteria were used for this human evaluation to ensure a standardized evaluation?

3. What is the correlation between the LLM's automatic evaluations and human judgments of query quality? How were discrepancies between the automatic scores (s_auto) and manual scores (s_human) addressed during the evaluation process?

4. Were different values of 'k' for hops around question entities considered to test the robustness of model responses in relation to variations in graph size and structure?

5. Can the authors provide a more detailed breakdown of performance across the different query types, specifically how various perturbation methods affected certain query types?

6. In section 4.2, it is unclear how the retrieved structured knowledge is integrated into the PLM. In particular, what are the differences in final sequence length across different forms of information (this can be essential for managing the LLM inference costs)?

7. Please explain why, in certain query categories (such as "evaluation and reflection"), the knowledge-augmented methods result in degraded performance compared to the baseline.

---

> ### Author Response · Authors · 2024-11-23
>
> We sincerely thank the reviewer for the thoughtful comments. Below, we address each concern in detail:
>
> ---
> ### **W1: How do the proposed G-retrieval and G-generator methods fundamentally differ from existing approaches like RAG and KG-augmented generation.**
>
> We would like to clarify that the primary objective of our work is not to propose state-of-the-art methods for close-ended or open-ended KGQA tasks but to establish a comprehensive benchmark to evaluate how well LLMs+KGs approaches perform under open-ended KGQA scenarios. The G-retrieval and G-generator methods are designed as simple baselines to represent two common workflows: (1) retrieving essential information from knowledge graphs and (2) subsequently leveraging that information to enhance the reasoning capabilities of LLMs during the generation process. We intentionally kept these methods straightforward to ensure clear ablation studies, focusing on whether integrating KGs can make LLMs more trustworthy.
>
> ---
> ### **W2 & Q1,2,3: How well do the generated queries authentically represent real-world questions, and how robust and reliable is the evaluation process, including the alignment between automatic and human assessments?**
>
> As mentioned in Appendix (Lines 861–894), we employed a **human-in-the-loop process** to ensure the generated queries authentically reflect real-world scenarios. The queries were refined iteratively based on human feedback, ensuring alignment with real-world use cases such as answering user-specific questions or conducting exploratory queries in knowledge graphs. The human feedback is reflected over multiple metrics, including naturalness, relevance, specificity, novelty, and actionability, to emulate characteristics of practical queries.
>
> For the human evaluation process, three evaluators participated in the manual assessment of query quality. All evaluators are computer science majors with fluent English skills. Since the evaluation focused on linguistic metrics such as naturalness, relevance, specificity, novelty, and actionability, we required only a fundamental understanding of English, without restricting the evaluators by their academic background. To ensure consistency, a detailed rubric and illustrative examples were provided to guide the evaluation process. Each evaluator worked independently to ensure unbiased assessments.
>
> For a sub-question “what is the correlation between the LLM’s automatic evaluations and human judgements of query quality? And how were discrepancies between the automatic scores ($s_auto$) and manual scores ($s_human$) addressed during the evaluation process?
>
> We reported the score distribution agreement between the people and LLMs (we use pearson correlation coefficient to measure the agreement between human and LLM's labels) as follows:
>
> | Metric | Round 1 | Round 2 | Round 3 | Round 4 |
> |---|:---:|---|---|---|
> | naturalness | 0.60 | 0.65 | 0.69 | 0.74 |
> | relevance | 0.55 | 0.59 | 0.64 | 0.70 |
> | specificity | 0.46 | 0.54 | 0.60 | 0.65 |
> | novelty | 0.49 | 0.57 | 0.63 | 0.67 |
> | actionability | 0.33 | 0.41 | 0.48 | 0.53 |
>
> It shows that as the number of human-in-loop rounds increases, the agreement between people and LLMs becomes closer.
>
> ---
> ### **W3: How do the four perturbation heuristics in OKGQA-P sufficiently represent the range of inaccuracies found in real-world knowledge graphs?**
>
> We appreciate the reviewer’s concern regarding the scope of perturbation heuristics in OKGQA-p. Identifying inaccuracies in real-world KGs is inherently complex, as they stem from diverse sources like link prediction errors, entity mislinking, and incomplete annotations. To address this, our benchmark focuses on **link errors**, which prior studies identify as the most prevalent and impactful type of KG inaccuracy.
>
> The four perturbation heuristics—relation swapping, replacement, rewiring, and deletion—were carefully selected to represent key manifestations of link errors:
>
> * **Relation swapping** simulates misclassified or mislabeled relationships.
> * **Replacement** introduces spurious links to emulate noise.
> * **Rewiring** reflects structural distortions in graph connectivity.
> * **Deletion** models missing edges or incomplete knowledge.
>
> These heuristics offer a practical and systematic foundation for studying the robustness of LLMs+KGs systems under common inaccuracies in KGs. While they may not capture every potential error type, they focus on the most statistically significant ones.

---

> > ### Author Response · Authors · 2024-11-23
> >
> > ---
> > ### **W4: how improvements in reducing hallucination relate to the retrieved KG knowledge?**
> >
> > We would like to highlight that in the experimental results presented in Table 2, we compare settings that utilize information extracted from KGs with those that do not rely on external knowledge (e.g., zero-shot or 4-shot prompting). The only difference between these settings is whether the prompt includes knowledge from KGs. Our findings indicate that incorporating knowledge from KGs generally improves performance on metrics such as SAFE and FactScore, which assess the hallucination ratio in the LLMs’ output.
> >
> > ---
> > ### **Q4: Did the study consider varying the value of ‘k’ (number of hops around question entities) to evaluate the robustness of the model’s responses against changes in graph size and structure?**
> >
> > We set K = 2 to balance graph size and computational feasibility, as increasing K leads to exponential growth in the number of edges and nodes, which can introduce excessive noise and make it more challenging to retrieve essential information from the KGs. This choice is also informed by common practices in other benchmarks, such as WebQSP and CWQ, where 2-hop subgraphs are widely used for similar KGQA tasks.
> >
> > ---
> > ### **Q5: Can the authors provide a more detailed breakdown of performance across the different query types, specifically how various perturbation methods affected certain query types?**
> >
> > Sure, we append this new experiment analysis in the revised version of our paper.
> >
> > ---
> > ### **Q6: How the retrieved structured knowledge is integrated into the PLM. In particular, what are the differences in final sequence length across different forms of information?**
> >
> > For the prompt construction, we follow the previous work KAPING[1]’s prompt template to transform different triplets extracted from KGs into a prompt. We provide statistics of the average token usages as follows for using different forms of information per questions of OKGQA.
> >
> > |  | triplets | paths | subgraphs |
> > |---|:---:|---|---|
> > | average token usage | 2,781 | 843 | 3,129 |
> >
> > * [1] Knowledge-Augmented Language Model Prompting for Zero-Shot Knowledge Graph Question Answering: https://arxiv.org/pdf/2306.04136
> >
> > ---
> > ### **Q7: Why in certain query categories (such as evaluation and reflection), the knowledge-augmented methods result in degraded performance compared to baseline?**
> >
> > We think these queries are in-generally much more complicated and require in-depth analysis. For example, a case in the evaluation & reflection category is “How do you evaluate Martin Luther King’s impact on the civil rights movement? Please explain your viewpoint”, the question demands not only retrieving relevant information about Martin Luther King and the civil rights movement but also synthesizing and analyzing that information to assess impact, consequences, and broader implications.
> >
> > In such cases, subgraph retrieval methods, which capture richer structural and relational information from the KG, are generally more effective. However, triplets, or paths retrieval may lack the depth needed to fully support this level of reasoning, leading to performance degradation compared to the baseline.

---

> > > ### Comment · Reviewer_k7xW · 2024-11-26
> > >
> > > Thanks for the response and for revising the paper. Regarding W4, my concern was more about the need to establish a clear link between the reduction in hallucinations and the specific piece of knowledge extracted from the KG during the generation process. This could contribute to the mechanistic interpretability of PLMs, as in [2, 3], though it is beyond the scope of the current work.
> > >
> > > [2] Unveiling the Black Box of PLMs with Semantic Anchors: Towards Interpretable Neural Semantic Parsing, AAAI 2023
> > >
> > > [3] Mechanistic Interpretability for AI Safety--A Review

---

> > > > ### Author Response · Authors · 2024-11-26
> > > >
> > > > Thank you for your insightful feedback and for raising this important point. The question of establishing a clear one-to-one mapping between hallucinations and specific pieces of knowledge is indeed open-ended. If we could achieve such precise mapping, it would signify a major breakthrough in understanding the limitations or knowledge gaps of LLMs (as your mentioned references). However, given the scope of our current work, it is challenging to conclusively determine this type of explicit relationship.
> > > >
> > > > We deeply appreciate you bringing up this question, as it opens up possibilities for extending our research. Specifically, it inspires us to explore whether for certain types of problems, it might be feasible to identify a strongly correlated set of knowledge that could fill the knowledge gaps of LLMs. We will carefully explore this direction in our future work.
> > > >
> > > > Additionally, we are eager to address any other concerns you may have. Since the current rating remains negative, we would appreciate it if you could let us know if there are any other matters or specific expectations that we could address to improve our rating. We are committed to doing our best to respond to your feedback. Thanks.

---

### Note · Authors · 2024-12-11

I have read and agree with the venue's withdrawal policy on behalf of myself and my co-authors.